# The Efficacious Benefit of 25-Hydroxy Vitamin D to Prevent COVID-19: An In-Silico Study Targeting SARS-CoV-2 Spike Protein

**DOI:** 10.3390/nu14234964

**Published:** 2022-11-23

**Authors:** Tomy Muringayil Joseph, Akshay Maniyeri Suresh, Debarshi Kar Mahapatra, Józef Haponiuk, Sabu Thomas

**Affiliations:** 1Department of Polymer Technology, Faculty of Chemistry, Gdańsk University of Technology, G. Narutowicza 11/12, 80-233 Gdańsk, Poland; 2Laboratory of Bacterial Genetics, Faculty of Chemistry, Gdansk University of Technology, 80-233 Gdańsk, Poland; 3Department of Pharmaceutical Chemistry, Dadasaheb Balpande College of Pharmacy, Nagpur 440037, Maharashtra, India; 4International and Inter-University Centre for Nanoscience and Nanotechnology, Mahatma Gandhi University, Kottayam 686560, Kerala, India

**Keywords:** 25-hydroxy vitamin D, Lopinavir, COVID-19, docking, simulation, Coronavirus

## Abstract

The environment has rapidly looked at proven specialist task forces in the aftermath of the COVID-19 pandemic to build public health policies and measures to mitigate the effects of emerging coronaviruses. According to the researchers, taking 10 μg of 25-hydroxy vitamin D daily is recommended to keep us safe. There have been several studies recently indicating that there is a reduced risk of contracting Coronavirus by 25-hydroxy vitamin D consumption, even though there is no scientific data to prove that one would not affect the COVID-19 viral infection by 25-hydroxy vitamin D consumption. In this regard, the present study investigates the important literature and the role of 25-hydroxy vitamin D to prevent COVID-19 infection by conducting an in-silico study with SARS-CoV-2 spike protein as a target. Lopinavir, a previously reported drug candidate, served as a reference standard for the study. MD simulations were carried out to improve predictions of receptor-ligand complexes which offer novelty and strength to the current study. MD simulation protocols were explored and subjected to 25-hydroxy vitamin D and a known drug, Lopinavir. Comparison of ligands at refined models to the crystal structure led to promising results. Appropriate timescale simulations have been used to understand the activation mechanism, the role of water networks for receptor function, and the ligand binding process. Furthermore, MD simulations in combination with free energy calculations have also been carried out for lead optimization, evaluation of ligand binding modes, and assessment of ligand selectivity. From the results, 25-hydroxy vitamin D was discovered to have the vital interaction and highest potency in LBE, lower RMSD, and lower inhibition intensity similar to the standard. The findings from the current study suggested that 25-hydroxy vitamin D would be more effective in treating COVID-19. Compared with Lopinavir, 25-hydroxy vitamin D had the most potent interaction with the putative binding sites of the SARS-CoV-2 spike protein of COVID-19.

## 1. Introduction

The COVID-19 (CoV) pandemic was first reported from Wuhan, the capital of Hubei province in China, with a series of mysterious and suspicious pneumonia cases, late in 2019 [1]. On 30 January, the World Health Organization (WHO) declared the Public Health Emergency of the International Convention (PHEIC), describing COVID-19 as an infectious disease on 11 March and as an epidemic on 11 February 2020 [2]. Since November 2002, a new strain of the Coronavirus, SARS-CoV, spread to 29 countries, declaring the first epidemic of a CoV infection which was reported from China, with clinical features including Severe Acute Respiratory Syndrome (SARS)-CoV [3], and the other in the Middle East was reported at first in 2012 [4] which is termed as Middle East Respiratory Syndrome (MERS)-CoV. The third epidemic, Coronavirus disease 2019 (COVID-19), is closely related to the SARS virus (Severe Acute Respiratory Syndrome Coronavirus 2 or SARS-CoV-2). The SARS-CoV-2 virus is responsible for the outbreak of corona disease in 2019–2020 and was first reported in Wuhan, China, and subsequently spread worldwide [5].

Supplementation with vitamins A, B, C, D, and E reportedly appears to have beneficial effects for patients suffering from COVID-19 [2,6,7,8,9]. Within such a panorama, it is worth mentioning that 25-hydroxy vitamin D mitigates the scope of acquired immunity and regenerates the endothelial lining. By evaluating the structure, it is obvious that 25-hydroxy vitamin D is a steroid hormone that can enter the nucleus and penetrate through membranes that make effective changes. In particular, the 25-hydroxy vitamin D receptor is a member of the nuclear receptor steroid hormone superfamily [10,11]. 25-hydroxy vitamin D passes across the membrane, influences the receptor’s binding, and goes into the nucleus, where transcriptional transition can be affected. In this aspect, it is not just 25-hydroxy vitamin D that needs to be replaced but an essential chemical that alters how the body behaves. 25-hydroxy vitamin D is much more complex, and the report shows that 25-hydroxy vitamin D is meant to regulate calcium [12]. 25-hydroxy vitamin D receptors have been seen in numerous cells, including the immune system [13]. The human body produces 25-hydroxy vitamin D through sunlight exposure or dietary supplements [14]. Reports show that individuals spending more time indoors in close isolation may also justify the spike in viral infections [15,16]. For more than a century, 25-hydroxy vitamin D deficiency has been indicated to improve vulnerability to infection. Kids with 25-hydroxy vitamin D deficiency suffer from nutritional rickets, and patients with kidney disease will lose 25-hydroxy vitamin D3 from their system and have chances of infections in the respiratory tract (RTIs) [17,18]. It has also been considered that 25-hydroxy vitamin D is essential for tuberculosis care [19,20]. Hence, RTI is a significant contributor to death, and 25-hydroxy vitamin D deficiency has become a major concern to public health. 25-hydroxy vitamin D levels can sometimes be higher for heart disease or cancer patients [21,22].

Subjects with COVID-19 showed leucocytes and lymphocytopenia, as well as a systemic elevation of pyrogenic cytokines such as interleukin (IL)-6, IL-10, and tumor necrosis factor (TNF)-α [6,9]. Several studies have reported an increase in neutrophilia and elevated D-dimer, as well as urea nitrogen (BUN) and creatinine in the blood plasma [6,9,10]. Increased interferon (IFN)-γ-induced-protein-10, plasma levels of IL-2, IL-7, IL-10, 10 kD monocyte chemoattractant protein-1, macrophage inflammatory protein 1-α, and granulocyte colony-stimulating factor were also reported [9]. 25-hydroxy vitamin D may decrease cytokine storm (cytokine release syndrome) and increase respiratory infection morbidity and mortality (Like COVID-19). A cytokine storm is a syndrome associated with acute respiratory distress (ARDS), a critical component of COVID-19 mortality. 25-hydroxy vitamin D also regulates or decreases pro-inflammatory cytokines by reducing the production of T helper cell inflammatory cytokines. The development of TNF-α and IFN-γ is therefore limited. Further evidence shows that SARS-CoV2 infections or COVID-19 infections are associated with the severity of this infection with cytokine elevation, IL-6.

Studies have shown some similarities between the biochemical aspect of what COVID-19 looks like and the biochemical aspect of what 25-hydroxy vitamin D deficiency looks like. Even though there is no evidence to prove this concept, it certainly raises eyebrows and began to look a little closer because it seems that TNF-α was elevated in both conditions, IFN-γ plus elevated late in the course for both COVID-19 and 25-hydroxy vitamin D deficiency. The same thing happened in the case of the Th1 adaptive response. Also, the decreased expression of ACE2 and hypercoagulability in both circumstances can be seen. All these correlations show that 25-hydroxy vitamin D may play a role in COVID-19 from all these observations. Studies also reported that smokers, especially older smokers, show higher densities of angiotensin-converting enzyme 2 (ACE2) receptors which cause severe COVID-19 pneumonia due to reduced tissue gene expression.

Knowing the 25-hydroxy vitamin D and antimicrobial pathway updates and their role in infection prevention is important. 25-hydroxy vitamin D upregulates cathelicidins and defensins, antimicrobial peptides, and the problem of antimicrobial peptides is, therefore, highly complex. Cathelicidin antimicrobial peptide (CAMP) and a single CAMP gene in humans concerning cathelicidins (hCAP18). In particular, 25-hydroxy vitamin D can boost the expression of the anti-microbial human cathelicidin peptide (hCAP-18), which is particularly important in host defenses against respiratory tract pathogens. It has been shown that respiratory-sensitive viruses activate these respiratory epithelial cells through the TLR3 receptor when they contact respiratory epithelial cells. This leads to activation of the 25-hydroxy vitamin D pathway, leading to increased regulation of the anti-microbial peptide CAMP or hCAMP. As far as protection is concerned, it can be expressed in leukocytes and epithelial cells. There are two alpha and beta protective classes, and it has been shown that the resistance bands to the influenza virus. In reality, they can induce influenza virus aggregation and decrease the ability of the virus to infect cells. The Antimicrobial peptides cathelicidins and defensins can also damage bacterial membranes. Not only do they participate in the pathway described here, but they also participate in the chemoattraction of different immune cells. Viral replication and the development of pro-inflammatory cytokines are decreased, which is essential for COVID-19 [23,24].

When we go to ethnicity, these darker skin races have raised the probability of death in COVID-19. Of course, mortality in Nordic countries is relatively poor. Yet their 25-hydroxy vitamin D deficiency, possibly due to widespread supplement use, is relatively rare. Hypertension, diabetes, obesity, and race have all been found to interact with 25-hydroxy vitamin D deficiency. The above characteristics are associated with an elevated risk of extreme COVID-19 [25]. Another thing we have lately discovered is the relationship between 25-hydroxy vitamin D and BMI. The more than 50 percent decline in obese subjects due to the availability of cutaneous synthesis of 25-hydroxy vitamin D3 could account for the clear finding that obesity is associated with 25-hydroxy vitamin D deficiency [26].

There is currently no international consensus on the optimum 25-hydroxy vitamin D supplementation available [27]. The report shows that supplementation of 25-hydroxy vitamin D contributes to fracture reduction [28]. Research undertaken in Germany found that people with 25-hydroxy vitamin D levels higher than 50 had more remarkable respiratory mortality survival than those with less than 30 [29]. A study reported in the British medical journal brings them together to see whether 25-hydroxy vitamin D supplementation in non-COVID-19 patients was a meta-analysis of several different trials. The study aimed to see whether 25-hydroxy vitamin D supplementation increased mortality and looked at 25-hydroxy vitamin D supplementation with 25 randomized control tests. The study found that treating 25-hydroxy vitamin D reduced the risk of acute respiratory disorders. Reports from the study demonstrate that there have been clear benefits for people who are very low in 25-hydroxy vitamin D and those taking daily or weekly supplements without extra bolus doses [30]. Another study was conducted in Japan, looking at a randomized trial of 25-hydroxy vitamin D supplementation in school children to avoid seasonal influenza A. This was done about ten years ago when 334 school children were given either 1200 foreign units of 25-hydroxy vitamin D3 a day or given a placebo each of them, and by doing nasal swab antigen checking, the endpoint was searching for influenza. The study learned during the winter season that those participants had an 18.6 percent incidence of influenza A but they got supplementary 25-hydroxy vitamin D with a 10.8 percent prevalence of influenza A wears those who got the placebo. The gap between the two is just 7.8 percent, which converts into the amount required to deal with 13, which means that this action is very effective. So obviously, there are children who are not generally at risk of 25-hydroxy vitamin D deficiency in school children with 25-hydroxy vitamin D supplementation. Although, in this population, the frequency of influenza A has decreased [31].

Since 25-hydroxy vitamin D as a possible drug candidate for COVID-19 will be the subject of this article, it is crucial to know the reported case studies. Another objective of the study is to understand the expansion of 25-hydroxy vitamin D as a therapeutic agent and think about its practical use against COVID-19. Many researchers have been interested in looking at that agent, and many scientific experiments have been carried out [32]. It is essential to look back on the evidence of COVID-19 influenza chest infections and then at the evidence for 25-hydroxy vitamin D as a preventive therapeutic agent against COVID-19. To substantiate 25-hydroxy vitamin D as a drug candidate, it is significant to move on to real hospitalization cases and then build up from the hierarchy of evidence for 25-hydroxy vitamin D in COVID-19 to randomized control trials that are the positive standard [33,34].

According to two randomized controlled trials (RCTs), winter 25-hydroxy vitamin D supplementation has been shown to minimize the chance of infecting influenza [31,32]. Some other RCT reports from Japan revealed the limitations of designing such clinical trials. This includes patients who have been vaccinated against influenza, have not measured baseline 25-hydroxy vitamin D levels, and are not benefiting from the use of 25-hydroxy vitamin D [35]. However, two recent RCTs that included subjects with higher-than-average baseline 25-hydroxy vitamin D levels have also been reported [35,36]. Despite the need for additional research to support their findings, Gruber-Bzura et al. suggested that 25-hydroxy vitamin D should lower the risk of influenza [37]. Additionally, human immunodeficiency virus type 1 (HIV) infections have been linked to the possible advantages of 25-hydroxy vitamin D supplementation. Mansue et al. [38] reported preclinical studies to account for the treatment of peripheral blood mononuclear cells with 1,25 (OH) 2D inhibits decreased cell susceptibility to HIV infection by inhibiting viral entry, modulating the expression of CD4^+^ cell surface antigens, damping viral p24 production, and limiting monocyte proliferation. It is independently linked to the progression of HIV to more advanced stages when baseline 25-hydroxy vitamin D levels are lower than 32 ng/mL, which appears to support the possible advantages of 25-hydroxy vitamin D supplementation to HIV patients. Regarding the potential effects of 25-hydroxy vitamin D supplementation on COVID-19-infected patients, experimental reports indicate that 25-hydroxy vitamin D can reduce the risk of COVID-19, considering the outbreak in winter (lowest serum 25-hydroxy vitamin D levels). 25-hydroxy vitamin D deficiency can cause ARDS, and mortality increases with age and co-morbidity of chronic illness, and such cases are directly associated with lower 1,25 (OH) 2D concentration. It can be assumed that 25-hydroxy vitamin D supplementation and the corresponding increase in serum 25-hydroxy vitamin D levels above 50 ng/mL (125 nmol/L) may positively reduce the incidence and severity of various viral diseases, including COVID-19. This is based on the protective effects in subjects at risk of chronic diseases, such as cancers, cardiovascular disease (CVD), respiratory tract infections, diabetes mellitus, and hypertension [39]. Given the well-known deleterious consequences of malnutrition [40], and keeping in mind the peculiarities of the ICU setting, [41] planned a pragmatic protocol for early nutritional supplementation of non-ICU patients hospitalized for COVID-19 [42] confirmed gastrointestinal clinical and laboratory features in COVID-19 from case reports and retrospective clinical studies. Notably, the prevalence of hypovitaminosis D among elderly Italians is extremely high, with a wintertime peak [43]. It has already been reported that 25-hydroxy vitamin D protects mice against acute lung injury caused by lipopolysaccharide induction by inhibiting the renin-angiotensin pathway and angiopoietin (Ang)-2-Tie-2 signaling [44]. Malek Mahdavi further proposed that 25-hydroxy vitamin D supplementation could be a feasible treatment strategy to treat COVID-19- and ARDS-induced diseases [45,46]. It appears possible that 25-hydroxy vitamin D prophylaxis (without overdosing) may lessen the severity of illness brought on by COVID-19, especially in environments where hypovitaminosis D is common. This is despite the likelihood that any protective effect of 25-hydroxy vitamin D against COVID-19 is related to the suppression of cytokine response [47]. In addition, Marik et al. proposed in 2020 that hypovitaminosis D may contribute to the regional differences in the reported case fatality rate of COVID-19, suggesting that 25-hydroxy vitamin D treatment may lessen COVID-19 mortality. These results demonstrated that sustained 25-hydroxy vitamin D deprivation could activate the RAS, which leads to lung fibrosis. [48]. Additionally, hypovitaminosis D encourages the chronic activation of the renin-angiotensin system (RAS), which can result in chronic CVD and reduced lung function [49]. In recent research, Tsujino et al. (2019) found that 25-hydroxy vitamin D3 is activated in lung tissue and that this activation has a protective impact against experimental interstitial pneumonitis in both human cells and mouse models of bleomycin-induced interstitial pneumonitis. According to Martineau et al., regular oral 25-hydroxy vitamin D2/D3 intake, especially in people with 25-hydroxy vitamin D deficiency, is safe and has a protective effect against acute respiratory tract infection (up to 2000 IU/d without an extra bolus). Lymphopenia is one of the primary symptoms of severe COVID-19 infection, and 25-hydroxy vitamin D administration raises the peripheral CD4 + T lymphocyte count in HIV infection [50,51]. The above observation speculated that COVID-19 infection would be more likely to occur if ACE2 levels were increased by CVD or RAS-blocking medications. In particular, low 25-hydroxy vitamin D levels appear to increase the risk of thrombosis because 25-hydroxy vitamin D regulates the expression of multiple genes important for angiogenesis, differentiation, cellular proliferation, and apoptosis [52]. In this regard, the requirement to admit COVID-19 patients to the ICU will be reduced by the administration of a proper dosage of 25(OH) 25-hydroxy vitamin D [53]. The report shows that the only human cathelicidin to be hydrolyzed by proteinase 3 between an alanyl and a leucyl residue to create the antimicrobial peptide LL-3 and inhibit platelet aggregation, hence lowering the risk of thrombus formation, is hCAP-18. LL-37 can reduce the phosphorylation of Src kinase and Akt^Ser473^, decrease platelet spreading on immobilized fibrinogen, and inhibit P-selectin expression on platelets [54]. Through ACE2 receptors on the endothelium, COVID-19 may infect endothelial cells, causing endothelial dysfunction [55]. The malfunction of 25-hydroxy vitamin D-binding protein binding to the ligand for VDR on the endothelium may contribute to the induction of endothelial dysfunction by a deficiency of 1,25 (OH) 2D3, which cannot effectively act as a ligand for the 25-hydroxy vitamin D receptor (VDR). Additionally, TNF elevates interferon (IFN), which causes secondary endothelial dysfunction and raises the risk of thrombosis, coagulopathy, and endothelialitis.

The above findings confirm that 25-hydroxy vitamin D-deficient patients are more at risk of death [56]. Hypovitaminosis D may be linked to a higher risk of COVID-19 severity, providing additional evidence of the beneficial effects of 25-hydroxy vitamin D supplementation on the immunological response [57,58,59]. Unexpectedly, but interestingly, 25-hydroxy vitamin D insufficiency is common in Italy and Spain [60]. In addition to exposure to UVB rays, intensive 25-hydroxy vitamin D supplementation as a potential preventive could be considered, as we still lack proven and efficient COVID-19 treatments. 25-hydroxy vitamin D supplementation is consistent with the premium non nocere principle due to the good tolerability and safety of even large dosages of 25-hydroxy vitamin D. The peculiar behavior of COVID-19′s dissemination, as well as the variety of clinical presentations and consequences, may be explained by research on 25-hydroxy vitamin D levels and VDR gene variants [61]. This poses significant concerns regarding the potential for higher viral pathogenicity in the population, given the association between immune system dysfunction and obesity [62]. The pulmonary microenvironment (e.g., alveoli) may be compromised by increased adiposity, immune cell trafficking, and viral pathogenesis may contribute to a maladaptive cycle of local inflammation and secondary injury. The situation may be worsened by high blood pressure and diabetes mellitus, which are frequently associated with obesity [63,64]. Hyperinsulinemia lowers 25-hydroxy vitamin D status in people with type 2 diabetes by promoting sequestration into adipocytes, which reduces the plasma membrane negative charge between red blood cells, platelets, and endothelial cells, enhancing agglutination and thrombosis [65].

An increased prevalence of venous thromboembolism (VTE), a clinical factor linked to a worse prognosis, has been observed in patients with COVID-19, particularly in hypogonadal men with a more significant hereditary predisposition which raises concerns about the safety of testosterone treatment [66]. However, the risk of VTE in people receiving testosterone treatment is a relatively recent concern. Recent case-crossover research included 39,622 men, and 3110 (7.8%) of them were found to have hypogonadism. Testosterone replacement therapy was associated with a higher risk of VTE in men with (odds ratio 2.32) and without (odds ratio 2.02) hypogonadism [67]. According to a report, the probability of severe lung involvement in patients with COVID-19 is linked to male testosterone levels. For the reason that androgen levels fluctuate throughout a person’s life, testosterone may positively and negatively affect the COVID-19 infection process [12]. Early on, the immunosuppressive effects of testosterone could account for men’s higher susceptibility to infection, which led some to believe that ADT might play a preventive function. Conversely, late-onset hypogonadism may have a less immunosuppressive impact on the cytokine storm at the time of infection in elderly males who frequently get ARDS. Testosterone inhibits the immunological stimuli-induced release of pro-inflammatory cytokines such as TNF-α and IFN-α, which can be detected in the peripheral blood leukocytes and show a worsening of the systemic inflammatory response in hypogonadism patients [68]. These findings further support the hypothesis that 25-hydroxy vitamin D prevents the cytokine storm and subsequent ARDS that is commonly the cause of mortality in COVID-19 infection [69,70]. 25-hydroxy vitamin D supplementation can lower excessive IL-6 levels in diabetic mice, but a 25-hydroxy vitamin D deficit in HIV-infected patients is linked to higher levels of IL-6 [71,72].

There are no pharmacological cures for COVID-19 at the moment, and there has been an increased interest in repurposing currently accessible medications for immediate usage due to the urgent demand for effective therapies. In this regard, the present study searched for literature and clinical studies that provided information on Lopinavir’s effectiveness in treating COVID-19. From the above aspects of 25-hydroxy vitamin D for preventing COVID-19 infection, the current study attempted to substantiate its potential role by molecular docking study. With the support of case reports and case studies, we studied the Lopinavir drug standard and 25-hydroxy vitamin D for in silico designs by molecular docking. For further relevant studies tying 25-hydroxy vitamin D to COVID-19, we searched PubMed, Google Scholar, and recent reference lists. Due to the diversity of the results and study methods, quantitative synthesis has not been made possible; instead, we narratively offered our findings. Also, it is the first attempt to perform and report the molecular dynamic study of 25-hydroxy vitamin D against the crystal structure of the SARS-CoV-2 main protease (PDB ID: 6LU7).

## 2. Materials and Methods

### 2.1. Molecular Docking

The GLIDE module of Schrodinger Maestro version 10.4 (Schrödinger LLC, New York, NY, USA) was used to conduct docking studies. Even if the Van der Waals radii of non-polar atoms are scaled, glide docking investigates the idea of a rigid receptor and can be used to model a slight “give” in the receptor or ligand [73].

### 2.2. Determination of Active Sites

The amino acid residues found in the active site were identified using the protein-ligand identifier profiler (PLIP) web server (https://projects.biotec.tu-dresden.de/plip-web/plip, accessed on 26 September 2022). The Glide Grid and the docking results were constructed and analyzed using the amino acids found in the active site [74].

### 2.3. Protein Preparation

The crystal structure of the SARS-CoV-2 main protease (PDB ID: 6LU7) from the Protein Data Bank (PDB) was chosen and used as a model for protein structure. Schrödinger Maestro 10.4 has been used for protein preparation [75]. The files were downloaded and saved as 6LU7.pdb. The protein preparation wizard of the Schrödinger protein structure was processed for its bond orders, alleviating potential steric clashes, formal charges, assigning disulfide bonds, missing hydrogen atoms, and incomplete and terminal amide groups. Water molecules that were present 5 Ǻ beyond heteroatoms were removed. The OPLS 2005 force field and the default constraint of 0.3 Ǻ RMSD were used for energy minimization. A PLIP (protein-ligand identifier profiler) web-server was used to identify the binding pockets, and amino acid residues were placed and used to generate a receptor Grid using GLIDE.

### 2.4. Ligand Preparation

The Ligprep module was used to prepare the ligand structures (25-hydroxy vitamin D) retrieved from PubChem Database (PubChem CID: 5280795). Lopinavir was regarded as the gold standard medication for comparing thermodynamic properties. Ligands downloaded in the 2D structure were converted to 3D, considering all the parameters. LigPrep generates the least 3D energy structure with the proper chirality. Hydrogen was inserted, angles and bond length were standardized, and the probable tautomer was introduced at a pH of 7.0. The final step of LigPrep includes energy minimization of ligand structures using the OPLS_2005 force field [76].

### 2.5. Induced-Fit Molecular Docking

The molecular docking study was performed using the Schrödinger Maestro v.10.4 GLIDE module [77]. Previously, the prepared receptor grid and ligand were used for docking, and Extra-precision (XP) docking was adopted using the OPLS_2005 force field. Even if the Van der Waals radii of non-polar atoms are scaled, glide docking addresses the idea of a stiff receptor and can mimic a slight “give” in the receptor or ligand, excluding near-contact limitations.

### 2.6. Molecular Dynamics Simulation

The accurate prediction of a ligand-protein complex structure is important for computer-assisted drug development. The present study aimed to demonstrate how molecular dynamics (MD) simulation can be used to evaluate a docking pose predicted by a docking program. MD simulation equilibrates the system and removes the ligand from the predicted position if the predicted pose is not unstable in an aqueous environment. Additionally, the study looked into molecular dynamics simulation to monitor and evaluate the conformational behaviors of the atoms and molecules, which validate the outcomes of molecular docking. System builder from the Desmond application of the maestro platform was used to create a system that included a macromolecule-ligand complex with solvent for simulation purposes. The solvent was modeled using an orthorhombic box and the TIP3P technology. 150 mM of NaCl ions were applied to the water-filled container to neutralize the systems. Then, the model system was relaxed before the simulation using the Desmond program’s Molecular Dynamics tool, with a simulation time set up for 100 ns under the usual NTP (constant number of particles (N), temperature (T), and pressure (P)) condition. The simulation interactive diagram tool further graphically evaluated the simulation data, providing details of the protein-ligand complex properties during the simulation period. Data on bond interaction and root mean square distance (RMSD) were evaluated to validate our findings in virtual docking further.

## 3. Results

### 3.1. Active Site Determination

The SARS-CoV-2 Mpro is crucial for viruses during the maturity of the viral proteolytic process. It has been thought of as a possible target because it cleaves polyproteins to create functional viral proteins. SARS-CoV-2 Mpro’s activity will be hampered, and the spread of infection will be stopped by interacting with test chemicals at its active site. Chains A and B are present in the active site of SARS-CoV-2 Mpro, which creates a homodimer. A compound with the formula *n*-[(5-methylisoxazol-3-yl)carbonyl] alanyl-l-valyl-n1-((1r,2z)-4-(benzyloxy)-4-oxo -1-f[(3r)-2 3-yl]-methyl-but-2-enyl) 6LU7′s natural ligand is L-leucinamide. Figure 1 shows the active site, and Table 1 lists the residues of amino acids at the active site of SAR-CoV-2 Mpro.

### 3.2. Molecular Docking

The main protease (M pro) of SARS-CoV-2 is an important Coronavirus enzyme that plays a crucial role in facilitating viral replication and transcription, making it a desirable therapeutic target for SARS-CoV-2. The Protein Data Bank and PubChem Data Base were used to retrieve the crystal structures of the SARS-CoV-2 main protease (PDB: 6LU7) and 25-hydroxy vitamin D (Pub-Chem CID: 5280795). Before the protein preparation, all of the crystallographic water molecules from the complex were eliminated, and polar hydrogen was added. The disulfide bond was treated. Protein structure optimization and minimization employ the OPLS-2005 force field. The energy minimization of the prepared protein structure was stopped when the RMSD of 0.30 was reached. The LigPrep module was used to prepare the ligands. The force field OPLS-2005 was used to produce potential conformations or orientations at a PH7.02.0. A grid is used to display the composition and attributes of receptor molecules. A grid having a diameter of 20 was built from the ligand. The partial charge cut-off for grid generation is set at 0.25, and the van der Waal radius scaling factor is assumed to be 1.0. The Extra Precision scrolling method was used to conduct the docking investigation (XP Score). Based on the glide score, the best pose was chosen.

According to docking experiments, 25-hydroxy vitamin D interacts with the SARS-CoV-2 main protease to attach to the hydrophobic pocket of the protein by hydrogen bonding with the ASN-119 amino acid residue and Van Der Wall’s interaction. The results showed that 25-hydroxy vitamin D was capable of combining with Mpro, and the torsional energy was the lowest when compared with the reference standard drug used in this study. The hydroxyl group of 25-hydroxy vitamin D and the thiol group of Mpro Asparagine (ASN) played a key role in increasing 25-hydroxy vitamin D binding and stability with the main protease pocket by contributing to the formation of hydrogen bonds. Figure 2A shows the binding of 25-hydroxy vitamin D at the active site of SARS-CoV-2 Mpro, and Figure 2B represents the surface image of 25-hydroxy vitamin D at the binding pocket.

Hydrogen bonds (H-bonds) and hydrophobic contacts are the most common non-covalent interactions observed in protein–ligands complexes. H-bonds have an essential role in ligand binding, and their consideration in drug design is important because of their significant influence on drug specificity. Hydrophobic interactions are the key driving forces in drug-receptor interactions and are most frequent in high-efficiency ligands. In our analysis, H-bonds and hydrophobic contacts were monitored throughout the MD simulations to investigate the atomic-level interactions of 25-hydroxy vitamin D and Lopinavir with SARS-CoV-2 main protease. The formation and distortion of H-bonds and hydrophobic contacts in the protein-ligand systems over the 100 ns trajectories were evaluated. 2D structure depictions of 25-hydroxy vitamin D and Lopinavir with SARS-CoV-2 major protease are shown in Figure 3. The results reveal that 25-hydroxy vitamin D showed similar ΔG binding with the target protein as Lopinavir. 25-hydroxy vitamin D was involved in 1 hydrogen bond interaction with ASN-119, and Lopinanavir shows 1 π-stacking interaction with HIS-41 which stabilizes binding.

### 3.3. Molecular Dynamics Study

The MD simulation results allowed us better to understand the structural dynamics of the receptor-CoV-2 protease-ligand complex. The stability of the drug-protein complex in an explicit solvent system was tested for 100 ns using the Desmond module of the Schrodinger equation. Appendix A provide an overview of the total, kinetic, and potential energy evolutions during simulation time for the complex systems under study, including spike-25-hydroxy vitamin D-ACE2, and spike-Lopinavir-ACE2, respectively. RMSD values obtained for 25-hydroxy vitamin D and Lopinavir after the simulation are shown in Figure 4.

Small, globular proteins can tolerate RMSD values with a change of 1–3. Lopinavir was used as a trial benchmark for the previously mentioned medication candidate. The evidence from the RMSD for 25-hydroxy vitamin D and Lopinavir during the MD run indicates that the values fall within acceptable limits, and as a result, the simulation is equilibrated. Throughout the simulation, protein interactions with the ligand can be visualized. The average RMSD value was found to be less than 2 in both protein and ligand. A significant difference was observed between the average RMSD value of protein and ligand. The plot shows that these interactions can be summarized and grouped by type (Figure 5).

Hydrogen Bonds, Hydrophobic, Ionic, and Water Bridges are the four forms of protein-ligand interactions (or “contacts”). Each interaction type has further, more detailed subtypes that can be investigated using the “Simulation Interactions Diagram” panel. It is evident from the protein-ligand contact graph created during the MD run that 25-hydroxy vitamin D interacts with the Mpro active site. After the MD run, study frames were chosen from the MD trajectories, and binding energies were computed using MMGBSA. The binding energies of the protein-ligand complex were calculated using frames chosen at intervals of 5 ns. Standard deviation was calculated from the results to get the average binding energy. Table 2 lists elements that make up total binding energy. Figure 6 shows the potential antiviral activity of Lopinavir, the repurposed drug candidate for COVID-19.

## 4. Discussion

There is a critical need to find new medications with unique mechanisms to limit viral transmission and processing given the severity of the SARS-CoV-2 outbreak and the potential development of comparable coronaviruses from animal reservoirs to create future pandemics. Viral enzymes such as proteases are often regarded as the main targets in the quest for novel anti-coronaviral medications to stop the viral replication cycle. Due to changes that take place in the active region of the enzyme, this technique may, however, result in the selection of inhibitor-resistant strains [78], which may lead to the preponderance of drug-resistant variations in the viral population. This suggests that focusing on viral attachment processes may be a much safer method of overcoming anti-coronaviral medication resistance, especially in the absence of efficient vaccinations. To comprehend how proteins work, it is crucial to determine their precise molecular structures. This process is also a key step in many drug development initiatives. Unfortunately, it will be difficult to discover the experimental structures of many proteins since there aren’t any. Computer modeling is one method for resolving this issue. However, the projected structures will have flaws since the existing modeling techniques are not particularly reliable. New methods are therefore required to improve the structural accuracy of protein models. The COVID-19 drug’s mechanism of action is depicted in Figure 7. The graphic shows an overview of virus-cell interaction-based treatment approaches to treat SARS-CoV-2 infection. Strategies that target the host include RBD mimics and antibody fragments such as scFv. Strategies that target viruses include antibodies or antibody fragments like Fc. The ACE2-RBD interaction is suppressed in both situations, avoiding infection.

Proteins have been folded effectively to native-like structures using MD simulations, although this method is only applicable to short amino acid sequences. It would seem more practical to use MD refinement of homology models based on excellent templates, which will have a pretty high overall accuracy in secondary and tertiary structures. The experimental structure might not, however, be a global energy minimum of the force field model. In structure-based drug design, precise modeling of protein-ligand complexes is crucial. These structures have been the basis for MD simulations that have been extremely helpful in studies of receptor function and drug binding. The activation mechanism, the importance of water networks for receptor function, and the ligand binding process have all been studied using appropriate timeframe simulations. Additionally, lead optimization, analysis of ligand binding modes, and evaluation of ligand selectivity have all been demonstrated to benefit from the use of MD simulations in conjunction with free energy calculations.

Target protein Main Protease (Mpro) has been identified as a significant potential therapeutic target. The reproduction, transcription, and pathogenicity stages of the viral life cycle depend heavily on the primary protease of the SARS-CoV-2 virus. To test the possible binding affinity at the likely binding sites of SARS-CoV-2 main protease, we selected Lopinavir and 25-hydroxy vitamin D for the current study. Compared with the reference standard medicine used in this investigation, the results revealed that 25-hydroxy vitamin D was highly capable of interacting with Mpro and had the lowest torsional energy. By contributing to the formation of hydrogen bonds, the hydroxyl group of 25-hydroxy vitamin D and the thiol group of Mpro Asparagine (ASN) has been essential in improving 25-hydroxy vitamin D binding and stability with the major protease pocket. 25-hydroxy vitamin D seemed to have the strongest interaction similar to the standard Lopinavir, the highest efficacy in LBE, the lowest RMSD, and the lowest inhibition intensity Ki. The interactions of 25-hydroxy vitamin D with the pocket may suggest that it is a powerful inhibitor of COVID-19 SARS-CoV-2 main protease binding site, which would support its inclusion in the regimen for treating COVID-19 alone or in combination with other medications used for this purpose. The substances under study, Lopinavir, and 25-hydroxy vitamin D have side chains and cyclic structures. The effectiveness of these compounds in preventing viral invasion was evaluated by subjecting these structural elements to MD simulations to investigate their binding affinities and conformations to the targeted protein.

The length of a computer simulation relies on the resource’s availability as well as the simulation’s goal. Therefore, if the RMSD is found to be a straight line for 20 ns or 30–50 ns, we may infer that the system has attained convergence and the simulated structure is rather stable. Since a steady line has been seen during the manufacturing phase, we expect the findings of the current investigation are satisfactory. Equilibrium is attained in the RMSD at 20 ns. Additionally, the calculated averaged attributes from a trajectory, such as the free energy or RMSD, stay constant (within an acceptable error). If the value of the property differs noticeably, keep lengthening the simulation period until convergence is attained, for instance by calculating the property of interest from simulations lasting 50 and 100 ns. We anticipate that 100 ns will be sufficient for the current investigation as the property does not change considerably [79]. According to MD simulations, which showed that the two chemicals bind to the protein snugly with favorable MM-GBSA and ligand efficiency values, the stability of complexes with the highest-scoring poses was validated. Additionally, the examination of atomic interactions at protein-ligand interfaces revealed that the 25-hydroxy vitamin D molecules we suggested were able to maintain a network of hydrophobic and hydrogen bonds with the key protease residues of SARS-CoV-2 for the course of the 100 ns trajectory. These results imply that the suggested drugs could stop the SARS-CoV-2 spike glycoprotein from interacting with ACE2 and stop the virus from infecting host cells. Additionally, small molecular structures may be more effective in inhibiting spike-ACE2 interaction than peptide inhibitors in terms of metabolic stability, oral bioavailability, solubility, clearance, and cost of manufacture [80].

The global spread of the COVID-19 infectious outbreak poses a threat to the survival of the whole human species. It is crucial to take all steps, no matter how little, to combat this illness. Antiviral medications treat COVID-19 infection, although not all individuals respond to them equally. Particularly in people who already have chronic illnesses like diabetes and obesity, the disease advances aggressively. In addition to these individuals, it was discovered that the illness advanced slowly under particular clinical circumstances. The levels of 25-hydroxy vitamin D3 in those with COVID-19 infection were shown to decline as the illness became more severe. Additionally, it was shown that the 25-hydroxy vitamin D levels of patients who needed to be hospitalized for this infection were lower than those of those who did not. Supplements and tablets are the two methods to add 25-hydroxy vitamin D to diet. Food supplements such as fish oils, some types of mushrooms, and egg yolks, which are also included in red meat, are excellent sources of 25-hydroxy vitamin D. Even while the appropriate functioning of the immune system cannot be enhanced beyond normal limits by any specific vitamin, food, or supplement, a well-balanced diet will help assure it. Even when there isn’t a global outbreak, individuals still need to eat a well-balanced diet that includes foods high in 25-hydroxy vitamin D such as eggs and fatty fish because it’s difficult to acquire enough of it from food alone. Some morning cereals, margarine, and yogurt are fortified with 25-hydroxy vitamin D. However, prolonged exposure to sunshine can occasionally result in a 25-hydroxy vitamin D excess.

Our findings point to the potential role of 25-hydroxy vitamin D in the inhibition of the SARS-CoV-2 spike protein target, which may serve as a starting point for the future development of a COVID-19 therapeutic candidate. The findings also point to the potential for expediting the structure-based identification and optimization of a new COVID-19 therapeutic candidate by using simulation-refined models of protein-ligand complexes. Few similar works have been reported so far on 25-hydroxy vitamin D against the main protease (M^pro^) and RNA-dependent RNA polymerase (RdRP) [81], Mpro and PLpro [82,83], endoribonuclease Nsp15 [84], etc. with impressive docking scores of >−10 Kcal/mol. But molecular docking as well as no dynamic study has been yet reported so far for 25-hydroxy vitamin D against SARS-CoV-2. Since SARS-CoV-1 and SARS-CoV-2 have very similar binding modes to human ACE2, 25-hydroxy vitamin D may have antiviral effects against both of these CoVs as well as any future CoVs that originate from animal reservoirs to infect humans [85]. Further in-vitro and in-vivo research into this connection is warranted [86,87]. If there is a strong signal, further study should help in figuring out the best dosage and duration of treatment. For viruses, the SARS-CoV-2 Mpro is essential throughout the maturation of the viral proteolytic process. Due to the fact that it breaks down polyproteins to produce useful viral proteins, it has been considered a potential target that 25-hydroxy vitamin D can inhibit successfully [88,89].

## 5. Conclusions

The pervasive actions of 25-hydroxy vitamin D on many organ systems have raised many possible interactions between it and the mechanisms by which the SARS-CoV-2 virus infects humans. The current work evaluated the literature and the role of vitamin-D in preventing COVID-19 infectious diseases. To substantiate the conclusions of the literature, the present study also examines the importance of 25-hydroxy vitamin D by conducting an in-silico experiment using the SARS-CoV-2 spike protein as a target with the previously mentioned drug candidate Lopinavir as a reference. MD simulation results assisted in distinguishing the structural dynamics of receptor CoV-2 protease-ligand complex. From the results, 25-hydroxy vitamin D was discovered to have the most vital interaction and highest potency in LBE, lower RMSD, and lower inhibition intensity concerning the standard. The putative binding sites of the SARS-CoV-2 spike protein of COVID-19 had the highest interactions with 25-hydroxy vitamin D compared with Lopinavir. The findings revealed that 25-hydroxy vitamin D was highly capable of combining with Mpro, and the torsional energy was the lowest similar to the reference standard drug used in this study. Both, the hydroxyl group of 25-hydroxy vitamin D and the thiol group of Mpro Asparagine (ASN) contributed to the formation of hydrogen bonds, which increased 25-hydroxy vitamin D binding and stability with the main protease pocket. The specific interactions of 25-hydroxy vitamin D at the binding pocket suggest that it is a possible inhibitor of primary SARS-CoV-2 protease and may favor its inclusion in the COVID-19 treatment regimen, either intrinsically or by the combination of other medications used for this purpose. Besides, regular 25-hydroxy vitamin D supplementation is recommended by health organizations and local health authorities by consensus. 25-hydroxy vitamin D has also been found to respond to respiratory diseases; thus, the study proves that adequate 25-hydroxy vitamin D levels in the human body can be an excellent candidate to protect against COVID-19 during this pandemic. The addition of 25-hydroxy vitamin D to the COVID-19 treatment protocol may have a desired effect on viral replication inhibition, enhancing immunity and reducing the mortality rate. The study attempted to give a new dimension to the risk and outcomes of the enigma caused by COVID-19, and further in vitro and in vivo research on the role of 25-hydroxy vitamin D would be timely revealing and warranted.

## Figures and Tables

**Figure 1 nutrients-14-04964-f001:**
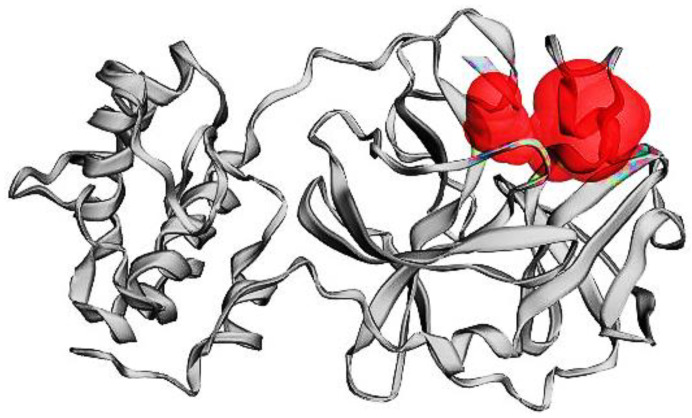
The active site of SARS-CoV-2 Mpro.

**Figure 2 nutrients-14-04964-f002:**
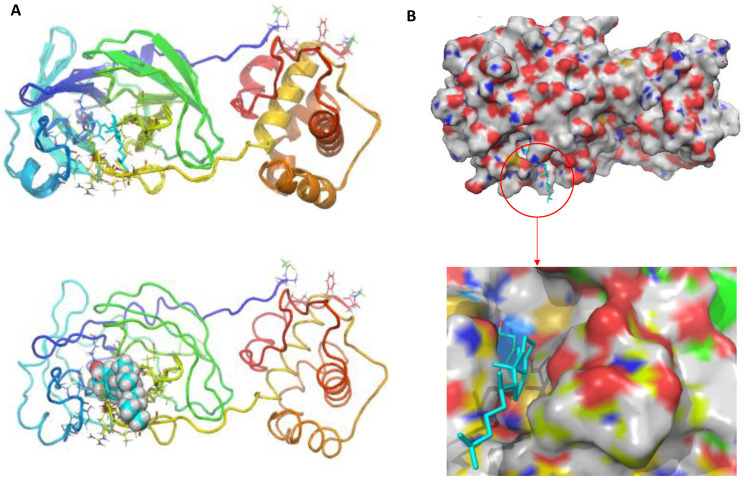
(**A**) The predicted binding mode of the 25-hydroxy vitamin D at the active site of SARS-CoV-2 major protease (**B**) Representation of surface image of 25-hydroxy vitamin D binding at the active site of SARS-CoV-2 major protease.

**Figure 3 nutrients-14-04964-f003:**
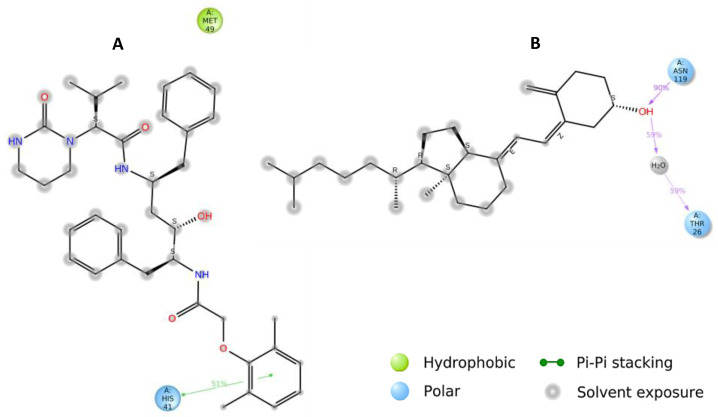
A schematic of detailed ligand atom interactions with the protein residues. Interactions that occur more than 30.0% of the simulation time in the selected trajectory (0.00 through 100.00 nsec), are shown. (**A**) Lopinavir (**B**) 25-hydroxy Vitamin D.

**Figure 4 nutrients-14-04964-f004:**
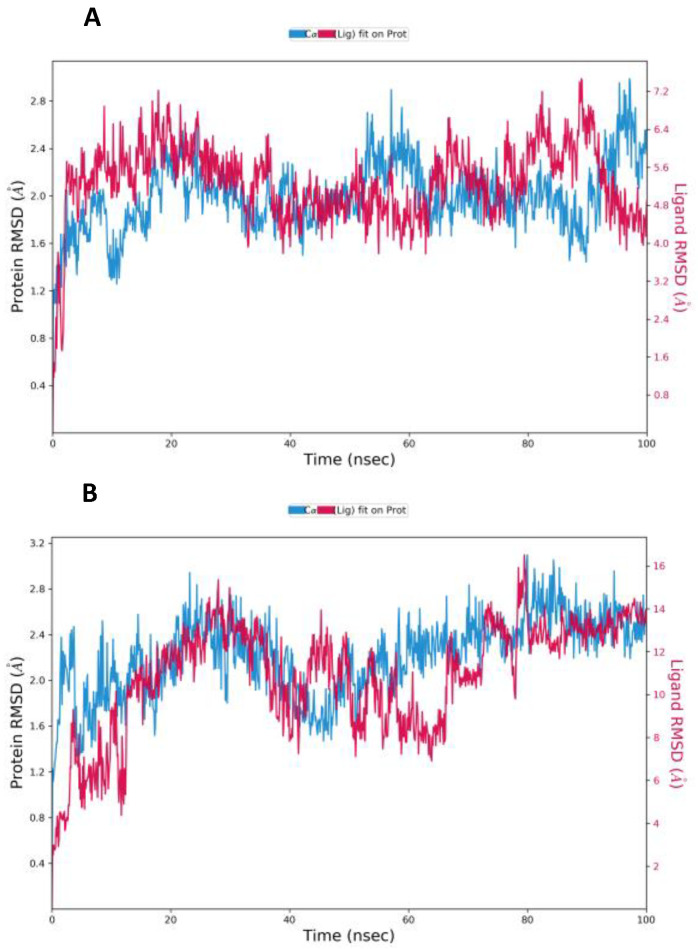
The root-mean-square deviation (RMSD) of the Cα atoms as a function of 100 ns simulation time for the ligands with Mpro (**A**) 25-hydroxy vitamin D (**B**) Lopinavir.

**Figure 5 nutrients-14-04964-f005:**
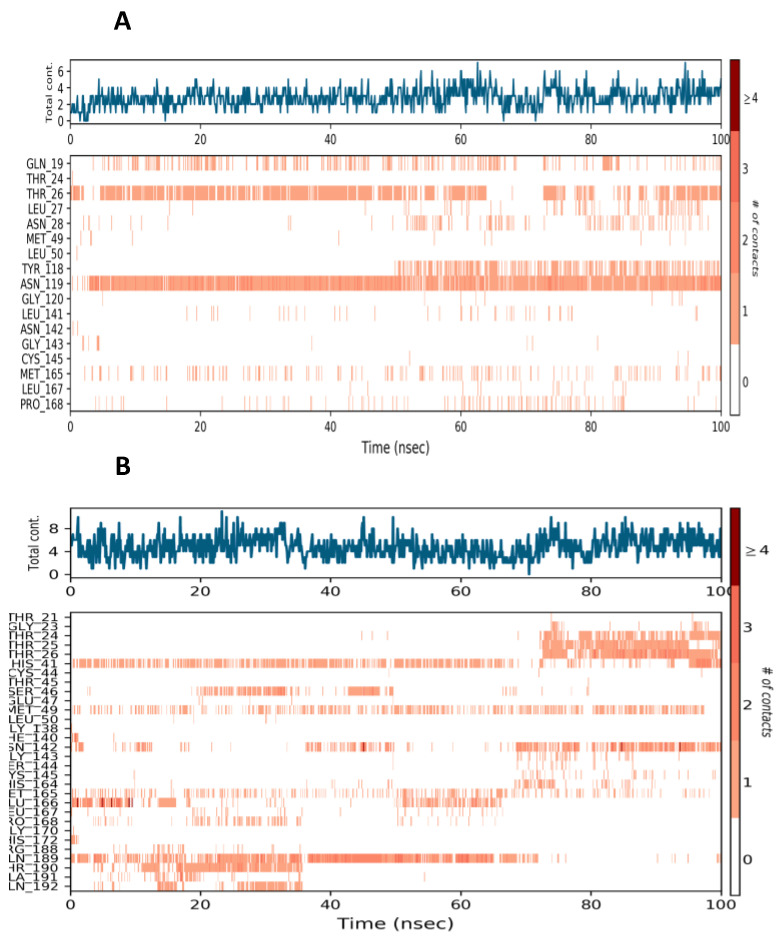
Protein-ligand interaction (**A**) Mpro-Lopinavir contacts at 30% consistency (**B**) Mpro-25-hydroxy vitamin D contacts at 30% consistency.

**Figure 6 nutrients-14-04964-f006:**
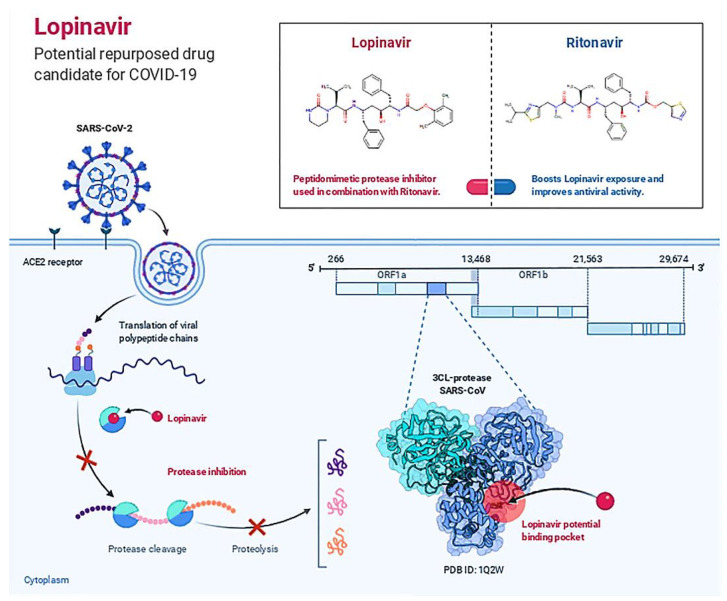
Lopinavir: Potential Repurposed Drug Candidate for COVID-19.

**Figure 7 nutrients-14-04964-f007:**
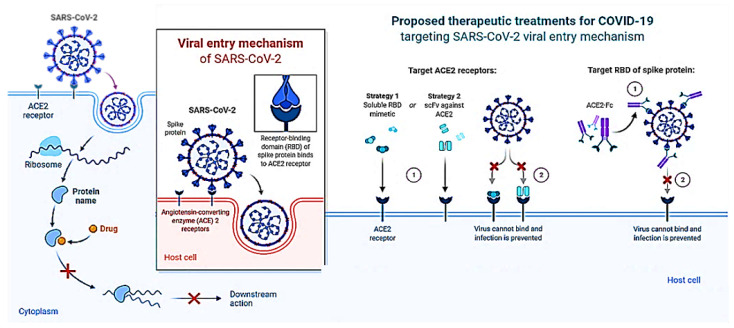
Schematic illustration of the COVID-19 Drug Mechanism of Action.

**Table 1 nutrients-14-04964-t001:** Active site residues of SARS-CoV-2 Mpro.

Residue	Amino Acid	Distance H-A	Distance D-A	Donor Angle
143A	GLY	2.00	2.87	145.90
144A	SER	3.65	3.99	104.01
163A	HIS	1.77	2.37	116.24
164A	HIS	1.85	2.80	161.75
166A	GLU	2.60	3.48	144.50

**Table 2 nutrients-14-04964-t002:** The contributions of various energy components to the total binding energy of the 25-hydroxy vitamin D-Mpro complexes *.

Drug	Average Binding Energy (kcal/mol)	Coulomb	vdW	Lipo	Bond	Packing	Solve GB
25-hydroxy vitamin D	−57.947	−12.462	33.426	−17.876	−0.815	0.00	27.28
Lopinavir	−58.193	−12.677	35.818	−18.394	−1.33	0.33	33.72

* Coulomb—Coulomb energy, vdW—Van der Waals energy, Lipo—Lipophilic energy, H-bond—Hydrogen bonding correction, Packing—pi-pi packing correction, Solv GB- Generalized Born electrostatic solvation energy).

## Data Availability

Not applicable.

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
