# Peer review of "The Efficacious Benefit of 25-Hydroxy Vitamin D to Prevent COVID-19: An In-Silico Study Targeting SARS-CoV-2 Spike Protein"

_nutrients, 2022, doi:10.3390/nu14234964_

Round 1
Reviewer 1 Report
1. The vitamin D should not capitalize vitamin D
2. To be accurate in the introduction the body does not produce vitamin D from dietary supplements it obtains vitamin D from supplements.
3. Unclear what is meant by vitamin D needs is derived from humans from sunlight penetration? Also unclear what is meant by you have a more significant potential for whole vitamin D? Clearly this manuscript needs very careful editing for English and syntax
4. To have a better insight regarding obesity and vitamin D status the authors should read and reference Ekwaru et al. PLOS one 2014. The effect of obesity on vitamin D status begins with a BMI greater than 30.
5. The authors used the term vitamin D levels when presumably they mean 25 hydroxyvitamin D levels or vitamin D status?
6. This is a review not a novel. They continually interject sentences like Here is a successful summary you can keep an eye on it.
7. In the conclusion it states there are 2 ways to get vitamin D into your diet. It would be better understood if they have said there are 2 sources of dietary vitamin D which of course is not correct. What is true is that you can obtain it naturally from some foods such as cod liver oil, oily fish like salmon and sun dried mushrooms. Mushrooms do not contain vitamin D without being exposed to sunlight. There is little vitamin D in red meat but there may be 25 hydroxyvitamin D3 which the authors should reference. In some countries such as India, United States, Canada foods are fortified with vitamin D including dairy. In India they also fortify cooking oil with vitamin D.
Author Response
We appreciate the time dedicated by the respected reviewer to read and comment on this manuscript. We modified the whole manuscript and the English language is now corrected and the coherence/link within the text is checked and improved. We also tried our best to improve the narrative structure by revisiting the introduction.
- This paper appears to be a review rather than a research paper because the introduction is too lengthy.
Taking note of the valuable comments, the introduction part is now rearranged with recent studies and the current state of the art. The introduction part clearly explains the overall research topic, as well as the depth of the information,was presented.The introduction part is now substantiated with proper the types of sources that will be included in the reference.The modified introduction part provides context by introducing the topic first: why is this topic interesting/significant, what do we know about it so far, how has the field progressed, and what new progress has been demonstrated etc. The newly added parts which are colored in blue were necessary to make the reader a better understanding.
- 2. I found a very similar paper published last year "Vitamin D is a New Promising Inhibitor to the Main Protease (Mpro) of COVID-19 by Molecular Docking". there is nothing new finding in the present study
As per the comment, we realize that a previous study shows a similar paper published last year entitled "Vitamin D is a New Promising Inhibitor to the Main Protease (Mpro) of COVID-19 by Molecular Docking".The study describes molecular docking used to calculate the binding energy, which is vital to interpret the biological activity of the ligands.
The accurate prediction of a ligand-protein complex structure is important for computer-assisted drug development. The goal of the present study was to demonstrate how molecular dynamics (MD) simulation can be used to evaluate a docking pose predicted by a docking program. MD simulation equilibrates the system and removes the ligand from the predicted position if the predicted pose is not unstable in an aqueous environment. Additionally, the study looked into molecular dynamics simulation to monitor and evaluate the conformational behaviors of the atoms and molecules, which validate the outcomes of molecular docking. MD simulation is also used to determine how a biomolecular system will respond to some perturbation.
In the present study, MD simulation of Vitamin D has shown greater efficacy in binding with Mpro of COVID-19 compared to the recently recommended drug. The docking study was simulated to streamline the interaction effects of Vitamin D and Lopinavir with the active site of Mpro. Vitamin D is found to have potential interaction in terms of total H-bond, van der Waal, torsional, and desolvation energy which were comparable to the standard drug, Lopinavir. In this regard, we hope, the molecular dynamics of Vitamin D with the Main Protease (Mpro) of COVID-19give more novelty and strength to the current study.
Our findings point to the potential role of vitamin D in the inhibition of the SARS-Cov-2 spike protein target, which may serve as a starting point for the future development of a COVID-19 therapeutic candidate. The findings also point to the potential for expediting the structure-based identification and optimization of a new COVID-19 therapeutic candidate by using simulation-refined models of protein-ligand complexes. Few similar works have been reported so far on vitamin D against main protease (Mpro) and RNA-dependent RNA polymerase (RdRP) [1], Mpro and PLpro [2,3], endori-bonuclease Nsp15 [4], etc. with impressive docking scores of > -10 Kcal/mol. But molecular docking as well as no dynamic study has been yet reported so far for vitamin D against SARS-CoV-2.
- Qayyum, S., Mohammad, T., Slominski, R.M., Hassan, M.I., Tuckey, R.C., Raman, C. and Slominski, A.T., 2021. Vitamin D and lumisterol novel metabolites can inhibit SARS-CoV-2 replication machinery enzymes. J. Physiol. Endocrinol. Metab. 2021.
- Tiwari, A., Singh, G., Choudhir, G., Motiwale, M., Joshi, N., Sharma, V., Srivastava, R.K., Sharma, S., Tutone, M. and Singour, P.K. Deciphering the Potential of Pre and Pro-Vitamin D of Mushrooms against Mpro and PLpro Proteases of COVID-19: An In Silico Approach. Molecules. 2022, 27(17), 5620.
- Al-Mazaideh, G.M., Shalayel, M.H., Al-Swailmi, F.K. and Aladaileh, S.H., 2021. Vitamin D is a New Promising Inhibitor to the Main Protease (Mpro) of COVID-19 by Molecular Docking. Pharm. Res. Int. 2021, 33(29B), 86-191.
- Shalayel, M.H., Al-Mazaideh, G.M., Aladaileh, S.H., Al-Swailmi, F.K. and Al-Thiabat, M.G., 2020. Vitamin D is a potential inhibitor of COVID-19: In silico molecular docking to the binding site of SARS-CoV-2 endoribonuclease Nsp15. J. Pharm. Sci. 2020, 33(5), 2179-2186.
- Figures 1-4 are only for explaining the introduction; they are not required in a research article.
Taking note of the valuable comments, the introduction part is rearranged with recent studies and the current state of the art.Figures 1-4 have been removed from the introduction part to make the introduction part stiffer and more precise.
4.The duration of the molecular dynamics study is only 20ns, which is insufficient to justify publication in high-impact journals nowadays.
Main Protease (Mpro) is a target protein that is reported as an important potential therapeutic target.The determination of detailed molecular structures of proteins is important to understand how they function and is an essential component of many drug discovery campaigns. Unfortunately, experimental structures are lacking for a large number of proteins and will be challenging to determine. One approach to solving this problem is to generate computer models. However, current modeling methods are not very accurate, and the predicted structures will contain errors. Consequently, there is a need for new approaches to increase the structural accuracy of protein models. In this work, we investigated the interaction of vitamin D with Main Protease (Mpro)of COVID-19 using molecular dynamics simulations which can improve predictions of the binding mode of a ligand. Our results suggest that simulation-refined models of protein-ligand complexes can contribute to accelerating structure-based discovery and optimization of a novel drug candidate for COVID-19.
Computer simulation time length depends on the availability of the resource as well as on the objective of doing the simulation. Thus, as per the suggestion, we repeated the study of the dynamics at 100ns. The results were promising and all the calculations based on the out put were incorporated.Besides, the averaged properties calculate from a trajectory, e.g., RMSD or free energyshowed positive variations. Since the property does not change significantly, we hope dynamics study at100 nsprovides more strength to the present study.
- overall, this manuscript does not provide any novelty
While the vaccines have provided adequate immunity against the virus, new variants, such as Omicron, with several mutations in the spike protein can be handled comprehensively by establishing a therapeutic strategy that will remain effective against future variants.Our data recommend a deeper investigation of around a dozen drugs for their potential to inhibit endosomal (catL)-mediated entry of the virus. Many effective vaccines are being applied worldwide, and they significantly reduce the number of new cases in countries with rapid vaccination rates. However, a treatment is still needed, since the vaccine protection is not 100%, and there is inequitable vaccine distribution across the world. Therefore, we believe our proposed regimen will help meet this need.The goal of this study was to assess if vitamin D could be a drug candidate for COVID-19. MD simulations were carried out to improve predictions of receptor-ligand complexes rather than a molecular docking study. MD simulation protocols were explored and subjected to vitamin D and a known drug, Lopinavir. Comparison of ligands at refined models to the crystal structure led to promising results. MD simulations show several advantages rather to a molecular docking study. First, it is unlikely that MD refinement improves the precision of docking, at the crystal structure during the simulation. Second, the best MD refinement protocol was able to refine a majority of the ligand binding modes and large improvements can be achieved in a few cases. Third, the MD model improved predictions of both the loop region and the binding mode of the ligand. Finally, the virtual screening performance of the receptor models could be improved by MD refinement.
MD simulations have been used successfully to fold proteins to native-like structures, but this approach is limited to short amino acid sequences. MD refinement of homology models based on good templates, which will have a relatively high overall accuracy in secondary and tertiary structures, may appear to be more feasible. However, it is also possible that the experimental structure is not a global energy minimum of the force field model. We understand that the accurate modeling of protein-ligand complexes is essential in structure-based drug design.
MD simulations based on these structures have proven very valuable in studies of both receptor function and drug binding. Appropriate timescale simulations have been used to understand the activation mechanism, the role of water networks for receptor function, and the ligand binding process. Furthermore, MD simulations in combination with free energy calculations have also been shown to be useful in lead optimization, evaluation of ligand binding modes, and assessment of ligand selectivity. Our results suggest that vitamin D has a positive effect on inhibiting SARS-Cov-2 spike protein target, which can guide further development as a repurposed drug candidate for COVID-19.Due to the diversity of the results and study methods, quantitative synthesis has not made possible; instead, we narratively offered our findings. Also, it is the first attempt to perform and report the molecular dynamic study of vitamin D against the crystal structure of the SARS-CoV-2 main protease (PDB ID: 6LU7).

Reviewer 2 Report
in this study entitled "Role of Vitamin D in the prevention of COVID-19; An in-silico 2 study with SARS-Cov-2 spike protein as a target" the authors did a molecular docking and molecular dynamics analysis of Vitamin D against the main protease of SARS-CoV-2. Since covid-19 is the most researched topic right now, the current work isn't very impressive and seems to be the same as several other works. There are many concerns, but the following are a few.
1. This paper appears to be a review rather than a research paper because the introduction is too lengthy.
2. i found a very similar paper published last year "Vitamin D is a New Promising Inhibitor to the Main Protease (Mpro) of COVID-19 by Molecular Docking". there is nothing new finding in the present study
3. Figures 1-4 are only for explaining the introduction; they are not required in a research article.
4.The duration of the molecular dynamics study is only 20ns, which is insufficient to justify publication in high-impact journals nowadays.
5. overall, this manuscript does not provide any novelty
Author Response
We value the reviewer's effort and time spent reading and providing feedback on this work. The entire document has been revised, and the English have been fixed. We have also enhanced the coherence and links within the text. Additionally, by going over the introduction again, we made every effort to strengthen the narrative framework. We also improved discussions as much as we could, where modifications are colored in yellow. Explanations to MD simulation have been written with appropriate descriptions which gives a better sense to the reader.
The first paragraph is now a better way to highlight the importance of the SARS-CoV-2 virus is responsible for the outbreak of corona disease in 2019–20, was first reported in Wuhan, China, and subsequently spread worldwide. The second paragraph deals with the importance of supplementation with vitamins A, B, C, D, and E which reportedly appear to have beneficial effects for patients suffering from COVID-19. It ended in Vitamin D deficiency which has been indicated to improve vulnerability to infection and become a major concern to public health. From the third to seventhparagraphs explain the studies which have shown some similarities between the biochemical aspect of COVID-19, Vitamin D, and antimicrobial pathway updates and their role in infection prevention.
In the meantime, we modified the language and now the meaning and spelling of words and sentences are clear. In the session to the introduction, we tried to emphasize the importance of Vitamin D supplementation and Vitamin D and COVID-19 Association through the case studies reported. We highlighted the fact that Vitamin D makes it possible to combat COVID-19 by showcasing the beneficial effects of vitamin D supplementation on the immunological response. We modified the structure of this introduction to some extent and also modified the language. From the literature review we realized that there are some gaps in this regard, for example, there is no invitro enzyme assay or in vivo study reported to prove that vitamin D could prevent COVID-19. Of note, we added recently published works on Vitamin D, and COVID-19.
In the discussion part, molecular docking and molecular dynamics analysis of Vitamin D against the main protease of SARS-CoV-2 has been explained. In the discussion, we corrected the text in terms of language and grammar. The discussion included some additional supplementary explanations to the ending part. We tried to improve the discussion as much as we could. Also, we tried to compare our results with similar systems in the literature and cited the papers in the discussion part.The conclusion part is now rewritten with the explanation of the present study and the general information from the conclusion part is removed.
Our findings point to the potential role of vitamin D in the inhibition of the SARS-Cov-2 spike protein target, which may serve as a starting point for the future development of a COVID-19 therapeutic candidate. The findings also point to the potential for expediting the structure-based identification and optimization of a new COVID-19 therapeutic candidate by using simulation-refined models of protein-ligand complexes. Few similar works have been reported so far on vitamin D against main protease (Mpro) and RNA-dependent RNA polymerase (RdRP) [1], Mpro and PLpro [2,3], endori-bonuclease Nsp15 [4], etc. with impressive docking scores of > -10 Kcal/mol. But molecular docking as well as no dynamic study has been yet reported so far for vitamin D against SARS-CoV-2.
- Qayyum, S., Mohammad, T., Slominski, R.M., Hassan, M.I., Tuckey, R.C., Raman, C. and Slominski, A.T., 2021. Vitamin D and lumisterol novel metabolites can inhibit SARS-CoV-2 replication machinery enzymes. J. Physiol. Endocrinol. Metab. 2021.
- Tiwari, A., Singh, G., Choudhir, G., Motiwale, M., Joshi, N., Sharma, V., Srivastava, R.K., Sharma, S., Tutone, M. and Singour, P.K. Deciphering the Potential of Pre and Pro-Vitamin D of Mushrooms against Mpro and PLpro Proteases of COVID-19: An In Silico Approach. Molecules. 2022, 27(17), 5620.
- Al-Mazaideh, G.M., Shalayel, M.H., Al-Swailmi, F.K. and Aladaileh, S.H., 2021. Vitamin D is a New Promising Inhibitor to the Main Protease (Mpro) of COVID-19 by Molecular Docking. Pharm. Res. Int. 2021, 33(29B), 86-191.
- Shalayel, M.H., Al-Mazaideh, G.M., Aladaileh, S.H., Al-Swailmi, F.K. and Al-Thiabat, M.G., 2020. Vitamin D is a potential inhibitor of COVID-19: In silico molecular docking to the binding site of SARS-CoV-2 endoribonuclease Nsp15. J. Pharm. Sci. 2020, 33(5), 2179-2186.

Reviewer 3 Report
Dear Authors,
Vitamin D and COVID-19: New Mechanistic and Therapeutic Insights
is a very well hypothesized project. Vitamin D is essential and plays key roles in several physiological systems of the body. The introduction, health benefits, molecular level interactions, molecular biological studies involving transcription factor activation data is reported.
The novelty is basically dependant on the insilico studies. A functional assay involving the protease mimic would have helped to solidify the hypothesis. But, the molecular modeling is essential for the making any biological assumptions.It will be good to provide a rationale with functional protease enzyme assay.
Author Response
Nil
Round 2
Reviewer 1 Report
1. The manuscript still needs careful editing for English and syntax
In the abstract it states COVID infection and vitamin D consumption. They have no evidence for how much vitamin D was consumed. they say in the abstract that those who are more likely to be outdoors are less likely to be infected suggesting that this is sun-induced increase in vitamin D status not vitamin D consumed.Presumably what they mean is that COVID infection was related to vitamin D status.
2. The authors inappropriately use the term vitamin D when they mean 1,25-dihydroxyvitamin D. There is no one hydroxyl group on vitamin D that they then claimed interacts with the receptor. This needs to be corrected throughout the manuscript.
3. they should not use the term vitamin D level since vitamin D was not measured. They should use the term 25-hydroxy vitamin D and its abbreviation throughout the manuscript for the term vitamin D status not vitamin D level
Author Response
- The manuscript still needs careful editing for English and syntax
- Thank you for the comment. We have now improved the language, corrected language issues, and removed typographical errors using GRAMMARLY software.
In the abstract it states COVID infection and vitamin D consumption. They have no evidence for how much vitamin D was consumed. They say in the abstract that those who are more likely to be outdoors are less likely to be infected suggesting that this is sun-induced increase in vitamin D status not vitamin D consumed. Presumably what they mean is that COVID infection was related to vitamin D status.
- Thank you for the comment. We have now removed the statement and written factual details in the Abstract.
- The authors inappropriately use the term vitamin D when they mean 1,25-dihydroxyvitamin D. There is no one hydroxyl group on vitamin D that they then claimed interacts with the receptor. This needs to be corrected throughout the manuscript.
- Thank you for the comment. We have taken 25-hydroxy vitamin D for the study and not 1,25-hydroxy vitamin D.
- They should not use the term vitamin D level since vitamin D was not measured. They should use the term 25-hydroxy vitamin D and its abbreviation throughout the manuscript for the term vitamin D status not vitamin D level.
- Thank you for the comment. We have now replaced the term “vitamin D” with “25-hydroxy vitamin D”.
Reviewer 2 Report
There are still some worries that the MD results are not as good as indicated in figure 4, and the author should include average RMSD values to demonstrate the significant difference.
Still, the abstract is too long; it should be no more than 1-2 pages.
Overall, the authors have worked to improve the manuscript.
Author Response
There are still some worries that the MD results are not as good as indicated in figure 4, and the author should include average RMSD values to demonstrate the significant difference.
Thank you for the comment. The average RMSD value was found to be less than 2 in both protein and ligand. A significant difference was observed between the average RMSD value of protein and ligand.
Still, the abstract is too long; it should be no more than 1-2 pages.
Thank you for the comment. We have made the Abstract part short. Now, it is less than 1 page.